# Introducing Low-Cost Sensors into the Classroom Settings: Improving the Assessment in Agile Practices with Multimodal Learning Analytics

**DOI:** 10.3390/s19153291

**Published:** 2019-07-26

**Authors:** Hector Cornide-Reyes, René Noël, Fabián Riquelme, Matías Gajardo, Cristian Cechinel, Roberto Mac Lean, Carlos Becerra, Rodolfo Villarroel, Roberto Munoz

**Affiliations:** 1Departamento de Ingeniería Informática y Ciencias de la Computación, Universidad de Atacama, Atacama 1531772, Chile; 2Escuela de Ingeniería Informática, Pontificia Universidad Católica de Valparaíso, Valparaíso 2362807, Chile; 3Escuela de Ingeniería Civil Informática, Universidad de Valparaíso, Valparaíso 2362735, Chile; 4Centro de Ciências, Tecnologias e Saúde, Universidade Federal de Santa Catarina, Araranguá 88906072, Brazil

**Keywords:** collaboration, Multimodal Learning Analytics, Social Network Analysis, Collocated Collaboration Analytics

## Abstract

Currently, the improvement of core skills appears as one of the most significant educational challenges of this century. However, assessing the development of such skills is still a challenge in real classroom environments. In this context, Multimodal Learning Analysis techniques appear as an attractive alternative to complement the development and evaluation of core skills. This article presents an exploratory study that analyzes the collaboration and communication of students in a Software Engineering course, who perform a learning activity simulating Scrum with Lego^®^ bricks. Data from the Scrum process was captured, and multidirectional microphones were used in the retrospective ceremonies. Social network analysis techniques were applied, and a correlational analysis was carried out with all the registered information. The results obtained allowed the detection of important relationships and characteristics of the collaborative and Non-Collaborative groups, with productivity, effort, and predominant personality styles in the groups. From all the above, we can conclude that the Multimodal Learning Analysis techniques offer considerable feasibilities to support the process of skills development in students.

## 1. Introduction

One of the main educational challenges for the 21st century is the development of core skills, such as communication, collaboration, knowledge of information and communication technologies, and cultural or social competences [1]. Active learning methods, such as collaborative learning, present a solution alternative to facilitate a means for the development of this type of skills [2], but at the same time, a challenge for the evaluation is how to measure which students are effectively exercising and developing the required skills. It is at this point that new information and communication technologies, such as the introduction of low-cost sensors in the classroom, can play a fundamental role in providing quantitative information to professors and coaches in different fields [3].

Implementing low-cost solutions in the classroom is today a completely feasible alternative. Small and low-cost single-board computers (such as Raspberry Pi) and single-board microcontrollers (such as Arduino) provide a huge opportunity for the implementation of low-cost sensors [4]. These types of solutions would help the development of core skills through collaborative learning, because the teacher would be able to have a greater and better amount of data generated by the students [5]. The automation of data acquisition and analysis processes is another direct benefit in the incorporation of sensors into the classroom [6]. In this work we aim to step forward to an automated collaboration monitoring, to help teachers to facilitate collaborative learning. We use a ReSpeaker device, based on a Raspberry Pi board, which provides direction of arrival and voice activity detection, for less than US$ 50. In contrast, some of the embedded products with similar characteristics, but not oriented to education, cost above US$ 400.

Serious games (SG) are a form of active learning that contributes to the formation of skills [7]. An example of SG is Lego Serious Play^®^ (LSP) that constitutes a way to carry out collaborative learning through a structured methodology, which allows collaborative thinking reinforcing teamwork. LSP has been used as a basis for the development of learning activities in higher education [8]. It has also been used as a mechanism for the creation of teams, with satisfactory results in different higher education programs, such as Business Administration and Management [9,10], Computer Engineering [11], Design Engineering [12], Electronic Engineering [13], Industrial Engineering [14], Computer Engineering [15], and Systems Engineering [16].

Agile software development methodologies, such as Scrum [17] and Extreme Programming (XP) [18], represent this paradigm shift: instead of detailing how to perform and document technical tasks, they focus on communication and collaboration among the participants of the development process. The adoption of agile practices has allowed an increase in the success rates in projects and improved the perception of quality in them. However, the formation of agile teams represents a tremendous challenge for Software Engineering education. There is a gap between what is taught in the classroom and the software development industry needs [19,20]. There are criticisms from academia and industry regarding the individual nature of Software Engineering courses [21], emphasizing the need for the development of communication skills [22], interaction and collaboration [23]. From the same discipline it has been identified that collaboration, effective communication, and teamwork skills are the most important [24]. An example of an initiative to use collaborative learning through LSP for the development of key skills in agile methodologies is Lego4Scrum [25]. Despite these advances, the practical evaluation and development of skills such as communication and collaboration remains a challenge. In this context, Multimodal Learning Analytics (MMLA) techniques have been used to investigate and evaluate the collaboration of individuals in learning environments [3,26], including those related to programming tasks [27], and in specific agile practices such as writing user stories [28].

Multimodal Learning Analytics methods can be very helpful in this context as they allow gathered learning traces from different sources (both manually and automatically) to be integrated and analyzed to obtain a more panoramic comprehension of learning processes [29].

In this context, we present the results of the introduction of low-cost sensors as a complement for the evaluation of the performance of an educational activity, using Lego^®^ bricks, in an undergraduate course of Software Engineering. This activity seeks to promote the development of communication and collaboration skills for the agile software development with Scrum. We explore the relationship between the performance of teams from the perspective of Software Engineering, with oral communication and collaboration among the participants of the work teams. To analyze the collaboration, we have used low-cost multidirectional microphones (based on Raspberry Pi) and Social Network Analysis (SNA).

The article continues as follows. In Section 2 a literature review regarding the use of Multimodal Learning Analytics for the learning of collaborative teams is introduced. Section 3 is devoted to explaining in detail the different instruments and methodological aspects used in the case study. In Section 4 the main results of the experiments are presented. Finally, Section 5 discusses the results obtained and presents the main conclusions of this work.

## 2. Related Work

The main concept behind collaborative learning is that knowledge can be built by means of group interactions, and that learning may be promoted when people get together with a common goal (e.g., study a given subject, acquire new skills, solve a given problem, or develop a project) [30]. Collaborative learning fosters a communicative atmosphere among students, normally allowing them to freely express themselves, and thus facilitating tutorship tasks towards individuals [31]. According to [32], the adoption of a collaborative learning approach allows one to shift from a teacher-centered education to a more student-centered one, where the teacher plays the role of a mediator instead of the traditional role of knowledge transmitter. Moreover, socially cooperative and interactive contexts may help to stimulate the development of psychological functions that characterize the human being [33] such as interpersonal abilities, effective communication, and leadership [34].

At the same time that collaborative learning is largely disseminated, the research literature is still paving the way for the construction of a more robust and replicable corpus of evidence about the impact of this approach in the many possible educational contexts and scenarios [35,36]. This mainly has to do with the inherent challenges of tracking an infinite variety of learning settings where the limits and boundaries are not always necessarily well defined and standardized. For instance, collaboration can occur at different levels, varying from simple forms of basic collaboration (informal and sporadic) to more complex ones (formal and continuous) [37], and collaborative learning can take place at different scales (small groups, pairs, an entire class), using different forms of interaction (virtual, blended, face-to-face) and different periods of duration (a few minutes, one class, several months) [30]. It is expected that the success of a given collaborative learning setting may be influenced by all these different possible configurations and nuances that may be considered during its learning design.

The access to low-cost sensors integrated to a new generation of emerging technologies with multimodal interfaces have helped the development of a new field of research called Multimodal Learning Analytics. MMLA deals with data collected and integrated from different and non-traditional sources, allowing a more panoramic understanding of the learning processes and the different dimensions related to learning [26,35,38]. According to [39], MMLA allows the observation of interactions and nuances that are frequently overlooked by traditional Learning Analytics methods, since the latter frequently exclusively depend on computer-based data. In this direction, the introduction of low-cost sensors allowed several MMLA studies that were not possible to easily conduct until recently. For instance, low-cost sensors were already used to track and measure students motion to evaluate their level of attention in the classroom [40], to estimate their skills in seminars [41] and to classify their postures in oral presentations [6,42]. Moreover, low-cost sensors were also used to register and detect students’ voiced interventions during the resolution of collaborative work activities [28,43].

Collaboration has emerged as one of the most prominent topics across several investigations in the MMLA arena [3] already presenting several interesting results, receiving proposals of architectures and standardizations for better collecting data emanating from these scenarios [44,45], as well as suggestions towards the integration of such kind of data with the ones normally collected through traditional learning systems [46]. Investigations of MMLA focused on collaboration have significantly increased in recent years; here we mention just a few to illustrate some of the existing (and sometimes preliminary) findings in the field. In [47], researchers collected logs from student interactions with tangible interfaces together with their gestures, and extracted meaningful indicators to predict learning gains. The authors were able to categorize student postures during collaboration into three main classes called active, semi-active, and passive, and have found that “active” postures were correlated with learning gains while passive postures were negatively correlated with learning gains. The authors also observed that students who are not familiar with the subject matter tend to maintain a larger distance with their peer during a collaborative task. A number of investigations related to collaboration and MMLA have also been developed under the scope of the PELARS Project (Practice-based Experiential Learning Analytics Research And Support) (http://www.pelars-project.eu/). For instance, Spikol et al. [36] recorded students during collaborative work focused on solving specific problems (prototyping an interactive toy, color-sorting machine, developing an autonomous automobile), and extracted several features from these recordings (distance between learners, distance between hands, hand motion speed, audio level, counting of the faces looking at a screen). Together with information related to the physical position of the blocks used in the project, the authors were able to correlate the distance between the hands of the students and the distance between learners with the quality of the student performance in learning activities. Moreover, Scherer et al. [48] focused on discovering low-level indicators (from audio and writing modalities) for classifying the expertise and leadership of peers during collaboration. Features such as voice quality and number of prosodic measurements (uninterrupted speech, pause duration, articulation rate, among others) together with writing-based features were used to identify socially dominant leaders and experts within a study group solving mathematical problems. Among other things, the authors have identified that the peak slope (a measure for the identification of breathy to tense voice qualities) is significantly different between leaders and other students (indicating that the voice quality of leaders is generally more tense then the voice of the others).

To the best of our knowledge, few works have focused on the use of MMLA to measure and understand collaboration in the context of software development and software engineering skills. Perhaps the most related subject to software development and engineering in the context of collaboration that has gained attention from the MMLA community is computer programming. In this context, Reilly et al. [49] have collected multimodal data from dyad collaboration (corporal gestures together with speech data) during the programming of a robot to solve a series of mazes using Tinker (a block-based programming language). The authors showed that some movements and patterns of gestures correlated with collaboration and learning gains. For instance, code quality correlated with increased movements of the right elbow, right shoulder, mid-spine, and neck, and overall collaboration strongly correlated with higher average talk time by pairs. Specifically, in the context of software engineering and agile methods, Noel et al. [28] explored the collaborative writing of user stories using multimodal data. The authors derived SNA metrics related to collaboration from audio recordings collected while students wrote user stories, and linked this data with human-annotated information. The authors have found that non-collaborative behavior is more observed in those groups where students presented lower productivity and were less professionally experienced in software requirements, and that teams with fewer interventions tended to produce a greater number of user stories.

Agile methods are an important subject in the software development industry and part of the fundamentals of Software Engineering courses at universities. Agile methods present an inherent nature for this kind of investigation considering that communication and collaboration skills are key to achieving good performance at work with this type of methodology [50,51].

More recently, the field encompassing studies focusing on describing knowledge gained through systematic studies of data has emanated from collaboration networks (normally multimodal data) and has been coined “collocated collaboration analytics” [3]. The experiments described in the present paper could be classified into the categories of MMLA and Collocated Collaboration Analytics.

## 3. Materials and Methods

The following sections present the details of the case study according to the guidelines proposed by Runeson and Höst [52].

### 3.1. Definition of Goals

The objects of study are teams of students studying the subject of Software Engineering. The Software Engineering course is currently in the seventh semester of a Civil Engineering degree in Computer Science at the Pontifical Catholic University of Chile (PUCV). Software Engineering is an obligatory subject, corresponding to the area of Applied Engineering, which provides the student with an integrated vision of software development. Predominantly, the Software Engineering course has been taught using a traditional methodology focused on the masterclass. The syllabus of the course considers the unit of “Agile Methods for Software Development”. Today the industry maintains a high demand for professionals who possess knowledge and skills in these subjects. The main skills [50,53,54,55] required in these topics are collaboration, effective communication, teamwork, self-management, and commitment. The development of this type of skill is a grand challenge, and several strategies and innovations have been emerging that try to help solve this problem [56,57]. In this scenario, MMLA techniques are emerging as a real alternative to using technology to measure and evaluate these skills. The main goal of the study is to evaluate the group performance in terms of productivity, the process performance, and the communication and collaboration of the team members. From the research perspective, student interactions and collaborations are explored while simulating the Scrum framework with the Lego4Scrum game. As secondary goals, the following are proposed:To define a method that allows the conducting of Scrum retrospectives facilitating the interaction between the members of a team.To record and analyze the individual and group behavior of the work carried out by agile teams.Evaluate the effectiveness of MMLA as a technique to measure and evaluate skills in agile teams.

### 3.2. Case Description

#### 3.2.1. Training and Learning Activities

To achieve the proposed goals, two 90-min preparatory working sessions were held, while the third session was 150 min long (as shown in Figure 1).

The goal of the first preparatory session was to train students in the estimation of effort for the construction of artifacts with Lego^®^ bricks. This session unfolded as follows. In the first 50 min, the professor addressed the topics related to the estimation of effort in agile projects. Specifically, the topics of Story Point and Planning Poker as an estimation technique were addressed. After this, a practical activity of 30 min was developed. The students were asked to form groups of five participants. The material delivered was a set of Planning Poker cards for each member and a bag with 40 Lego blocks of different sizes. Each team had to estimate the time needed to build a tower with the Lego^®^ bricks delivered. The towers should be stable and as high as possible. The activity with the Lego^®^ bricks was very entertaining for the students and allowed the students to acquire some experience in estimating effort in the construction of objects using Lego^®^ bricks. It also allowed the training of the Planning Poker technique, where the students had to agree on estimates. At the end of the session, the results obtained by the teams were evaluated and important aspects of the experience were analyzed (see Figure 2 and Figure 3).

The second preparatory session had the goal of training the students with the Scrum framework. The students were grouped into teams of five participants and were given the following materials: five user stories, a 25 × 25 cm Lego flat base and 100 assorted Lego^®^ bricks. This session served as a pilot, allowing researchers to test the instruments, tools, and work areas for the teams. As the students were already familiar with the general concepts of agile methods, the professor used 40 min to review the contents of the Scrum framework. A working guideline for the development of the retrospective was designed and the objective of this ceremony was explained to the students. The challenge of the activity consisted of the construction of artifacts that would allow an astronaut to live on the planet Mars.

#### 3.2.2. Case Study

After the two preparatory sessions, the case study was carried out. Students who participated were 16 and were grouped into four teams. The activity was designed based on the proposal defined by Lego4 Scrum [25], and was adjusted according to the needs of the research. The activity lasted approximately 150 min, divided into the following steps (see Figure 4): 1. Instruction and organization of teams (20 min), 2. DiSC^®^ test [58] application (5 min), 3. Product backlog (15 min), 4. Sprint Planning (10 min), 5. Sprint 1 (10 min), 6. Retrospective 1 (20 min), 7. Sprint 2 (10 min), 8. Retrospective 2 (20 min) , 9. Sprint 3 (10 min), 10. Retrospective 3 (20 min), 11. Survey of satisfaction, evaluation and opinion (10 min).

We used an online chronometer to manage the time spent in each activity (this chronometer was projected to all students). During steps 4, 6, 8, and 10 of the activity, the students were recorded using ReSpeaker devices, a 4-microphone expansion board for Raspberry Pi. These microphones were used to record and store the students’ speech interventions, identifying the opportunities in which they spoke, as well as the duration of each intervention carried out. For the preprocessing, analysis, and reporting of the results, software generated by the authors was used [5]. During steps 5, 7, and 9 of the activity, students were videotaped through a tablet that recorded the whole process of building artifacts with Lego^®^ bricks in each sprint.

An interesting aspect to explore in the study is about the possible relationships that may exist between behavioral styles and collaborative work. For this reason, each student answered the 26-question DiSC^®^ test. DiSC^®^ is an evaluation instrument that measures the behavior and emotions of people based on four dimensions of personality. The DiSC^®^ test provides a natural profile of the individual, i.e., the innate way of doing things. It also provides an adapted profile, i.e., how it faces what happens in the environment. The DiSC^®^ methodology is based on the work of psychologist William Marston [58], and divides into 4 the different types of personalities, as shown in Table 1.

The DiSC^®^ tests were tabulated for each student and the natural and adapted profiles were determined. For each Scrum ceremony, worksheets were prepared that the students had to complete with data. These data allowed a quantitative record of the work done by the students in each group. The information obtained from these records allowed us to compare the way in which the teams planned and prioritized their work, the estimated effort levels and the degree of compliance with the planned work. These results are shown in Section 4.

For the researchers, the design of the retrospective was a key point because it should stimulate dialogue between the members of the team. For this, the proposal of Derby & Larsen [24] was considered, which indicates that a five-step agenda should be prepared for each ceremony, which are: (Step 1) Set the Stage, (Step 2) Get data, (Step 3) Generate ideas, (Step 4) Decide what to do, and (Step 5) Close the retrospective. For each of these stages, various facilitation techniques are selected depending on the objective to be achieved, the project status, the team’s state of mind, and the productivity achieved, among other factors. For the purpose of the investigation, the following restrictions were considered: 1. The technique must be done with the seated members; 2. The time for the retrospective must be 20 min; and 3. Only the work guides should be used as support, so there is no intervention of coach or Scrum Masters during the execution. Therefore, the proposal of [24] was modified and the agenda was reduced from five to three steps: (Step 1) Set the stage; (Step 2) Get data, generate ideas and decide what to do; and (Step 3) Closing of the retrospective. Table 2 shows the retrospectives activities agenda. The techniques were selected considering the restrictions of the experiment to be carried out and according to expert coach recommendations.

The survey for students collected different aspects related to the level of involvement and effort, the degree of participation, motivation, improvement in the learning process and teamwork. The student had to self-assess according to the mentioned items on a Likert scale from one to five (Strongly disagree—In disagreement—Neutral—In agreement—Totally agree). The items of the students’ survey were:Do you think that this type of activity helps to understand the benefits of teamwork?Do you think that working with Lego^®^ bricks facilitated collaborative work?Do you think working with Lego^®^ bricks facilitated communication among team members?Were all team members were actively involved?Were you according to how to plan and distribute work?Has planning poker helped the team reach consensus?Did you actively participate in the planning?Was there clarity in what each one should do?Have all team members worked?Did you finish each task assigned to you?Were problems correctly detected?Was it possible to adjust the work to achieve the objective?

Other open questions of the survey were:What I liked most was:What I liked the least was:As a team, what we would do differently would be:At the end of the activity I feel:According to work done, I can say that the leadership style that predominates in the team was:What do you think was the team’s greatest strength?What do you think was the team’s biggest weakness?

To determine the validity of the surveys, the technique of the judges was applied, submitting said surveys to the review of the researchers so that they could assess the pertinence of the items. Their indications were taken into account, and their formulation was made by consensus.

#### 3.2.3. Materials and Instruments

For the case study, a workspace was prepared for each team, where two trapezoidal tables were put together and four chairs were placed around them. An important aspect to consider in this preparation was the distance between work teams to avoid interference in the capture of audio with the microphones. Figure 5 shows the workspace prepared for each team. Along with the above, at each table the following materials were available:1Kit Lego^®^ Classic Creative 10,696 (484 pieces)1Lego^®^ flat base 10,701 (38 × 38 cm)1ReSpeaker Microphone1Tablet Samsung Galaxy Tab A6 10’4Set Planning Poker13User Stories (for details see Table 3)Guidelines and guides for the work:Prioritized backlog, Planned vs. done, guides for retrospectives, Fun vs. utility.Markers

#### 3.2.4. Research Questions

According to Robson’s classification, the objective of the study is exploratory, so that its findings can help to generate new hypotheses for future research [59]. The scenario of the case study is a real-world Software Engineering course, and the data collection and analysis is both quantitative (performance measurements, collaboration, contribution, and working documents) and qualitative (DiSC^®^ test, surveys, notes of field, and working documents).

We define the goal of the study in terms of [60] guidelines. The object of the study are agile development teams, in the context of undergraduate students performing a simulation of three development sprints using Lego^®^ as development materials of artifacts required in terms of user stories. The purpose of the study is to evaluate the group performance in terms of productivity (delivered functionality in terms of User Story Points), the process performance (ability of the group to consistently produce better estimations for the work to be done in the next sprints), and the communication and collaboration of the team members (based on the communication dynamics of the group, automatically measured with ReSpeakers). The perspective is from the point of view of researchers in education technology, in order to explore relationships between communication and collaboration and performance measurements, and the relationships of leaderships and personality characteristics with communication and collaboration dynamics. The quality focus is in the group conformance and communication and collaboration dynamics that could allow teachers to better implement the collaborative learning activity supported by low-cost sensors.

We propose three research questions: the first two relate to the performance of teams, to illuminate the key aspects of collaboration that can impact in the effective implementation of Scrum retrospective dynamics in the classroom, and the third question is aimed at identifying which leadership and personality characteristics have to be considered to conform collaborative groups:

The research questions defined are:RQ1: How does collaboration and communication relate to the productivity of agile teams?RQ2: How does collaboration and communication relate to the estimation of complexity of each sprint?RQ3: What is the relationship between the leadership and personality characteristics of agile team members, and the collaboration in during the activity?

To answer these research questions, we consider the following data sources:The ReSpeaker record file of the interventions made by each participant in each Scrum retrospective. We will collect speaking time and number of intervention metrics, to measure communication, and we will analyze the collaboration type by processing these measures (see Section 3.4).The worksheets were delivered, completed by each team as they progressed in the Scrum simulation; we will obtain the planned and accomplished story points and user stories, which will allow us to measure the quality of the development process in terms of the burning down of stories and the debt of user stories and story points in each sprint.The DiSC^®^ behavior test [58] that each participant answered anonymously at the beginning of the activity. Only the team number was tagged to triangulate the information. We will characterize each team by the most repeated personality type.The leadership test [61], completed by each participant, about the leadership style of the team during the activity. We will characterize each team by the most repeated leadership type.

#### 3.2.5. Ethical Considerations

All students were asked to complete an informed consent form to ensure the confidentiality of the data obtained. All the participants signed in and were considering in the results.

### 3.3. Definition of Metrics

For this study, the following metrics were defined:Productivity—Story Points Delivered (SP-D): In the planning game, each team defines the story points for each user story, as a complexity estimation. Subjects must report, in each sprint, the Delivered Story Points (SP-D). Productivity is measured as the number of accomplished story points; to compare it across teams, we will transform the number of story points to the percentage of story points produced in the sprint with respect to the total story points produced.Complexity Estimation—Story Points Debt (SPDebt): Subjects must report in each sprint, the Planed Story Points (SP-P). We will measure the quality of estimation improvements in terms of the debt of Story Points (SPDebt); this is the difference between the planned and the delivered story points.Communication—Speaking Time and Number of Interventions: from ReSpeakers recordings of the three retrospective ceremonies (performed after each development sprint), we obtain the total Speaking Time (ST) for a sprint, by adding all the speaking times of all the team members. Also, we obtain the number of interventions of each team member, and add all the interventions to get the total Number of Interventions (NI) of the sprint. Although this metric is obviously simplistic to assess effective communication, it will be analyzed in the context of the Retrospective Ceremony structure, purpose, and results, i.e., we are not looking for results such as “Group 1 communicated better than Group 2 because Group 1 has more ST or NI”, but to explore if a greater ST or NI could relate to Process Productivity or Process Quality.Collaboration—Collaboration Type by Permanence (CTPer) and Prompting (CTProm): To characterize the collaboration between team members, we use the ReSpeaker data to perform Social Network Analysis and calculate Prompting and Permanence metrics, which yields to a collaboration type for each team. Prompting (calculated from the number of interactions of each team member) and prompting (calculated from the speaking time), were already used to measure collaboration in [5,28]. We labeled each sprint retrospective as Collaborative or Non-Collaborative. Also, we labeled the teams as collaborative if they are collaborative in two or more retrospectives, and non-collaborative in other cases.Predominant Personality Type (PPT): Using the DiSC^®^ survey results of each member, a group will be characterized by the most repeated personality type among its members. If there is not a predominant type, it will be labeled as ”undefined”Predominant Leadership Style (PLS): Using the leadership survey results of each member, a group will be characterized by the most repeated leadership style among its members. If there is not a predominant type, it will be labeled as ”undefined”

### 3.4. Collaboration Data Analysis

For the analysis of the data obtained in the retrospectives through the ReSpeaker devices, a previously developed MMLA software was used [5]. This software receives the data from the microphones that were processed by Raspberry Pi devices and stores them in a database. Subsequently, SNA techniques are used to process the data and generate visualizations. Firstly, each group can be seen as an influence graph [62] (V,E,w,f) with four nodes, where each node a∈V is a member of the team, and a directed edge (a,b)∈E represents the interventions of Student *a* that are replicated by Student *b*. The number of these interventions is given by the edge weight, which is denoted by w(a,b). Furthermore, each node a∈V has a label given by f(a) that represents the speaking time of Student *a*. The pauses of the senders are not considered in the total speaking time. With all the above, we consider the following two centrality measures, used to characterize collaborative groups [5,28]:The *permanence* of *i*: $Per\left(i\right)=f\left(i\right)/\sum_{j\in V}f\left(j\right)$, i.e., the speaking time of participant *i* regarding the total duration of the conversation.The *prompting* of *i*: $Pro\left(i\right)=\bar{w}\left(i\right)/\sum_{e\in E}w\left(e\right)$, where $\bar{w}\left(i\right)=\sum_{j\in V}w\left(i,j\right)$, i.e., the number of times the author received comments from another interlocutor, regarding the total number of interventions during the entire conversation.

## 4. Results and Discussion

### 4.1. Qualitative Results

For the teams that were studied, the behavior profiles allow us to visualize the expected work environment in each team. At the beginning, the students should show a behavior based on their *natural profile*, i.e., they will behave in front of the work in an innate way and as they usually do in their activities. This should change in times of stress, such as situations of low fulfillment of tasks in each sprint, problems of organization of the team, clutter at work, or exhaustion of equipment. In this type of situation, it is possible to anticipate the appearance of the *adapted profile* of the students, which could cause changes in the environment and dynamics of the team. When observing the *natural profile*, 67% of the students are grouped between the Steadiness and Compliance factors, which indicates that a passive and introverted personality predominates among the students. This situation increases to 80% when observing the *adapted profile*. In Table 4, it is possible to observe the number of students classified according to the resulting DiSC^®^ factor.

After answering the DiSC^®^ test, the Scrum simulation work with Lego^®^ bricks started. The first job was to build the “Product Backlog” from the user stories that were delivered as requirements. The students analyzed each of the user stories and estimated effort using the Planning Poker technique. The two previous preparatory sessions gave students the experience to make better estimates. Planning Poker facilitates the interaction of students because everyone must participate by indicating with a letter (story points) the level of difficulty they observe with each story user. This process was observed by the researchers, checking that the students adopted the technique, and it was applied without major problems. The values assigned by the groups are shown in Table 5. Team 4 is the one that has less variation among assigned story points, unlike teams 1 and 2, which show values well above the other groups.

Once the “Product Backlog” was defined by the groups, the ”Sprint Planning” ceremony was held. To do this, they used the prioritized backlog and the estimates were made. The challenge for the students was to distribute the work in the three planned sprints. The results observed in this ceremony indicate that the groups distributed the user stories equally in the sprints. The difference observed is the product of the variability in the estimate shown in Table 5.

### 4.2. Quantitative Results

The main results obtained from the ReSpeaker devices are shown in Figure 6, Figure 7 and Figure 8, and in Table 6 and Table 7. Figure 6, Figure 7 and Figure 8 illustrates the 4 groups considered in each sprint retrospective, represented as influence graphs. The size of each node *i* reflects its label f(i). The thickness of each edge *e* reflects its weight w(e).

Table 6 shows the speaking times for each member of each group, and for each sprint. From the total speaking time of each group in each sprint (seventh column), it is possible to calculate the percentage of speech of each individual within their group in each determined sprint. From these percentages, the percentile of each individual is determined, considering his percentage of speech within his group, but taking as a range of data the percentages of all the groups within that sprint. In this way, the percentile represents the speaking time of an individual, normalized by the speaking time of all the groups in that sprint. After this, the value of the percentile is truncated to a value between 1 and 3 (“classification” columns). Finally, the number of maximum values obtained in the classification columns is counted, which generates a value (#max) between 1 and 4. A group is considered collaborative (C) if this last value is greater than or equal to 2, i.e., if there are at least two individuals who dominated the conversation within the group (under the speaking time criterion). On the contrary, the group is non-collaborative if the resulting value is 1, i.e., if there is only a single leader who dominated the conversation to a large extent.

Now, if we change the permanence criterion, satisfied by the measurement of the speaking time, for the prompting criterion, favored by the number of interventions of each student, then we obtain new data as illustrated in Table 7.

Under this criterion, the only group that maintains its overall collaboration behavior between sprints is group 4. The biggest difference with respect to the previous data is that here, Student 1 does collaborate a lot in the first sprint, being the second with the highest number of interventions. This means that during this sprint, the student intervened on several occasions, but his interventions were mostly brief. In fact, it is observed that this group has the lowest number of interventions among the four, although it is not the group with the shortest speaking time. This could perhaps be due to the fact that the group was more dedicated to advancing in the manipulation of Lego^®^ bricks, showing greater efficiency in their dialogue, and thereby ensuring a better collaboration. All the remaining groups presented greater differences with respect to the previous criterion. In this case, Group 1 did work collaboratively during the first two sprints, although in the third sprint the annulment of Student 2 is again observed, as opposed to the domination of Student 1. Group 2, meanwhile, this time presented a collaborative behavior in the third sprint, due to the greater number of interventions of Student 4 (which also increased their speaking time, although not enough to lead the group’s collaboration under the permanence criterion). Group 3, on the other hand, did not show any collaboration during the whole activity. As the speaking time of Student 2 of Group 1 was decreasing through the sprints, in Group 3, Student 4 was progressively decreasing his number of interventions (in comparison with those of his classmates). From a dominant leader, Student 4 happened to intervene very little in the last sprint, and the group could not maintain a dialogue with at least two active leaders.

In general, it is observed that during the third sprint the speaking time as well as the number of interventions within the work teams decreased considerably. This, as has been demonstrated, does not imply a decrease in their collaboration capacity, but may be due to other causes, such as a feeling of tiredness of the participants, due to the time elapsed and to the fulfillment of the requested requirements, or because they finished the activity in a good way and have no more to say in the retrospective.

### 4.3. Analysis of the Research Questions

Considering the previous description and characterization of groups, we present an exploratory analysis of these variables. Due to the limited number of groups, we yield to conclusions through analytical induction, instead of statistical inference [63]. The data for each metric previously discussed, per group, is in Table 8. Data for each sprint is presented in Table 9.
RQ1: How does collaboration and communication relate to the productivity of agile teams? We theorize that a greater communication in retrospectives would yield to more productive sprints, this is, a greater improvement in productivity. In Figure 9 we represent the cumulative percentage of Speaking Time (ST) in each sprint, compared to the cumulative percentage of the total story points delivered in the sprint.

As the chart shows, for Sprint 1, Group 3 (G3) is the less productive and G4 the more productive. In the Retrospective Ceremony of the Sprint 1, G1 used less than a 20% of the total speaking time of the team, while G2 has the higher speaking time, and also, the better improvement of productivity. This leads us to think that a greater speaking time during the retrospective could yield a greater improvement, but this is not the case for G4, whose improvement is not greater than G3, with almost the same speaking time. However, it is important to consider that G4 was the more productive in Sprint 1, so improvement could be harder than for G3. Our hypothesis is *H1: For a similar productivity, teams which communicate more during the retrospective could yield to greater improvements in the productivity of the next sprint.* This is an interesting topic regarding collaborative learning in the classroom: the introduction of sensors could allow the tracking of the communication levels, enabling teachers to perform interventions to foster the communication in less communicative groups, and allowing them to focus on groups that really need mediation in this process.

Another analysis in relation to this is that for all groups, the first two retrospectives took nearly 80% of the total speaking time. This suggests that the final retrospective was less active; although it could be caused by tiredness, we propose two hypotheses: *H2.1: In the final retrospective, communication will decrease compared to the previous sprints, because teams do not need to coordinate the next sprint*, and *H2.2: In the final retrospective, communication will decrease compared to the previous sprints, because teams learn from the previous retrospectives, so they can collaborate better in subsequent retrospectives*. For testing this second hypothesis, in future studies a fourth sprint will need to be introduced.

This finding may allow teachers to better design collaborative activities regarding retrospectives in Scrum, assigning less time for the final retrospective and maximizing the time for collaborations that are more important for performance. However, further studies are needed to explore the reduction of communication across sprints.

Regarding collaboration and productivity, we theorize that a better collaboration in retrospectives would lead to a greater productivity and a better process performance, assuming better coordination, complexity estimation, and distribution of the work for the next sprint. Considering the data from Table 9, we characterized the four groups according Predominant Collaboration Type by Permanence, i.e., the most repeated collaboration type across the sprints for a group. Under this definition, Groups 1 and 2 are labeled as non-collaborative, and Groups 3 and 4 are labeled as collaborative. The boxplot presented in Figure 10 compares the Delivered Story Points in all the sprints for the collaborative teams (by prompting) and the non-collaborative teams. Each box is defined by six data points (one point per sprint, with two groups in each category). Although at first sight the results could be interpreted as “Non-Collaborative groups are more productive than collaborative groups”, it is important to consider that all the groups were given the same user stories, which were completed by all the groups, and that story points are estimated by each team in a collaborative activity (Planning Game). Given these facts, we hypothesize that *H3.1: Non-Collaborative groups make more pessimistic complexity estimations*, and, by looking at the dispersion measurements, *H3.2: The range of the estimations of Non-Collaborative groups is greater than the estimation of collaborative groups.* With these facts, although productivity could not be compared, results suggest that collaborative groups perform better in terms of the precision of the complexity estimations during the retrospective.

These findings could have interesting consequences for collaborative learning: by improving the collaboration in groups, retrospective ceremonies could lead to better agreements in the complexity estimation of story points’ convergent, consistent, and more effective complexity estimations, as is predicted in the literature. However, if the design and setup of the learning activity does not promote collaboration, the benefits of the ceremonies could not be visible to students, and they could not get to experience the richness of the teamwork in the SCRUM ceremonies.

RQ2: How does collaboration and communication relate to the estimation of complexity of each sprint?

We theorize that a process that improves its performance will tend to make more consistent estimations, considering a major knowledge in the activity, the material, the performance of the team in previous sprints, and in the communication and collaboration. By looking at Figure 11, we can see the behavior of the four groups. All the groups present a similar behavior, thus, a consistent reduction of the debt of story points (to make them comparable across groups, Planned and Delivered Story Points were transformed to a percentage of the final Delivered Story Points).

However, it is important to explore the deviations of each group according to their own complexity estimations and agreements. As Figure 12 shows, the debt G1 presents the higher variance in its debt (the higher debt in the first sprint, the lower in the third). G2 and G4 have exactly the same debt behavior (that is why G2 line is not depicted in the figure). G3 presents less variation in debt. Considering that all groups managed to finish all the user stories (and deliver all the story points), we can explain this behavior as the precision of the estimations of complexity. When related to Collaboration Type by Permanence, groups labeled as collaborative (G3 and G4) present a similar behavior; however, one of the Non-Collaborative groups (G2) has a similar behavior too, so no conclusions can be drawn regarding collaboration types.

When comparing Story Points Debt with the Accumulated Speaking time, we obtain the chart presented in Figure 13. Considering that the behavior of groups is very similar (four groups have approximately 40%, 40%, 20% of the speaking time distribution in Sprints 1, 2, and 3 respectively), no conclusions can be drawn among the relationships of Speaking Time and Story Points Debt.

RQ3: What is the relationship between leadership and personality characteristics of agile team members, and the collaboration during the activity?

Considering G1 and G2 as Predominantly Non-Collaborative and G3 and G4 as Predominantly Collaborative, according to the Collaboration types by permanence of the hours in each sprint (Table 9), the Predominant Leadership Style and Predominant Personality Type presented in Table 8, we can observe that collaborative groups also have a democratic leadership style. This allow us to hypothesize that *H4: Collaboration leads to a democratic leadership style in the group*, which, according to Goleman, is useful for building consensus through participation, and establishing a positive work climate [61]. However, in order to study this hypothesis, it is important to randomly assign subjects to groups, to avoid grouping by convenience and confounding collaboration dynamic with previous bindings among participants. This topic could be relevant to allow teachers to form groups based on leadership styles, ensuring that democratic leaders could scatter across groups.

## 5. Conclusions

Multimodal data provides a holistic image of learners and the learning process [4]. Under this perspective, low-cost sensors, such as those used in this work, are presented as a promising alternative. However, there are constraints, such as ambient noise or mobility, which makes it hard to operate the sensors in outdoor or noisy areas [5]. In this exploratory study, we focused on the analysis of MMLA as a complement to the evaluation of complex skills such as collaboration and communication. This analysis was made by simulating the Scrum framework using Lego bricks for the construction of artifacts. The Scrum simulation was designed based on the Lego4Scrum guide, modifying elements according to the research objectives. Regarding the achievement of the defined objectives, we can conclude that they have been satisfactorily achieved by allowing the definition of new hypotheses that will surely set the tone for our future works. The MMLA software [5] and the application of techniques of SNA allowed the characterization of the study groups as Collaborative or Non-Collaborative. The qualitative data were analyzed and correlated with the quantitative data, allowing the discovery of interesting relationships, such as the prevailing DISC^®^ [58] factors with the story points estimates and the dominant leadership styles [61] with productivity.

Regarding RQ1, the retrospectives were designed to be the communication and collaboration instances par excellence. It was possible to observe that effectively the groups concentrated a large part of the communication in the first two retrospectives, reflecting the above in an improvement in the productivity of the team. Groups 1 and 2 are labeled as Non-Collaborative, and Groups 3 and 4 are labeled as collaborative. Group 4 was the one that showed the best characteristics related to collaboration, which is why it was recognized as the group that better reflected in its work the advantages that collaborative teams have and their impact on productivity. Therefore, it is interesting to be able to make more measurements to evaluate if it is valid to infer that more communication recorded in the retrospectives delivers greater productivity to the teams. Students who communicate more should be more productive.

Regarding the findings when analyzing RQ2, it is possible to point out that the groups indicated as collaborative showed a lower variability in the estimates of story points. This is related to what is analyzed in RQ1, in the sense that a group with a collaborative nature will generate greater and better communication. The foregoing is presumed to influence the results of the estimation of construction complexity. A greater discussion of complex issues diminishes the team’s perception of difficulty.

Regarding the analysis of RQ3, we found that the collaborative groups (G3 and G4) recognized that during the development of their work, a democratic type of leadership prevailed, as defined by [61]. Now, if we analyze the behaviors delivered by the DiSC^®^ test [58], we find that the Steadiness factor predominates in G4, which favors collaborative work. For the learning activity, the students formed a group by affinity, so it is interesting to test this type of activities by making different configurations of leadership behaviors and styles.

From the field notes and survey, we can conclude that there was a sense of joy in the classroom. We observed a fairly positive atmosphere during the learning activity and the students were very concentrated in each group. One thing that does not fail to attract attention is the scarce use of mobile phones, which reflects levels of commitment, happiness, and concentration of the students during the activity.

In light of the findings, future work will be oriented towards the design of replicable experimental studies to test our initial findings. Explicit groupings of subjects considering DISC and leadership styles will allow us to test its relationship with collaboration. We are also working in the online processing of the collaboration metrics to be able to automatically collect and display collaboration measurements, which will allow us to test the progression of the collaboration and also to provide a systematic approach for the replication of our measurements.

In view of this first exploratory study, we feel aware of the need and motivation to continue deepening the issue, and with the implementation of new technologies in future studies, we can establish the best comparative studies. In addition, we believe that methodological improvements can be made to obtain a greater quantity and quality of data for the use of MMLA techniques.

## Figures and Tables

**Figure 1 sensors-19-03291-f001:**
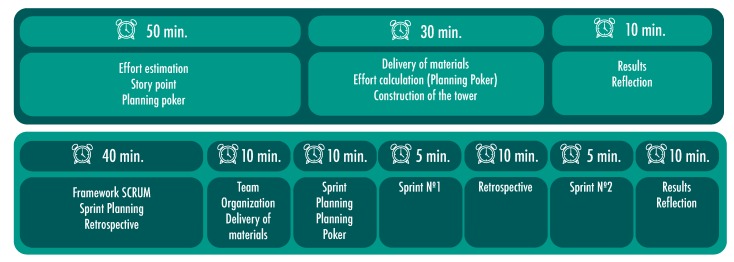
Planning of the two preparatory sessions.

**Figure 2 sensors-19-03291-f002:**
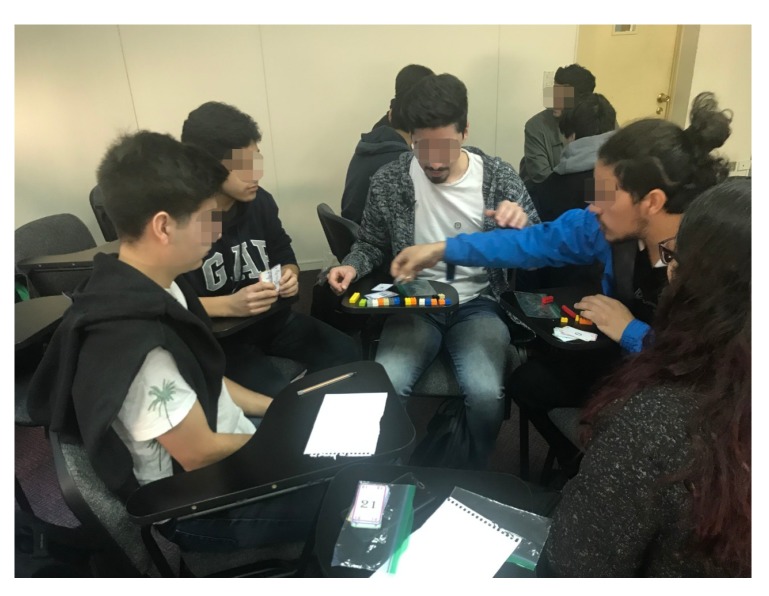
Students working.

**Figure 3 sensors-19-03291-f003:**
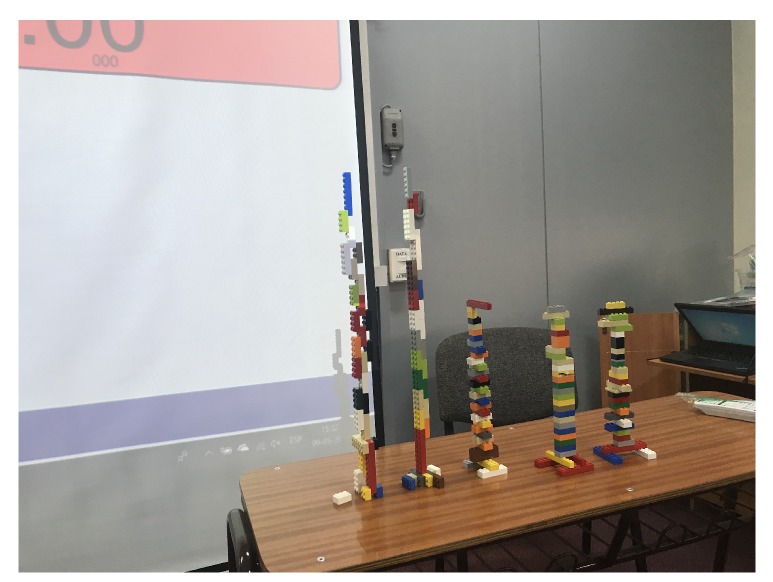
Some results obtained.

**Figure 4 sensors-19-03291-f004:**
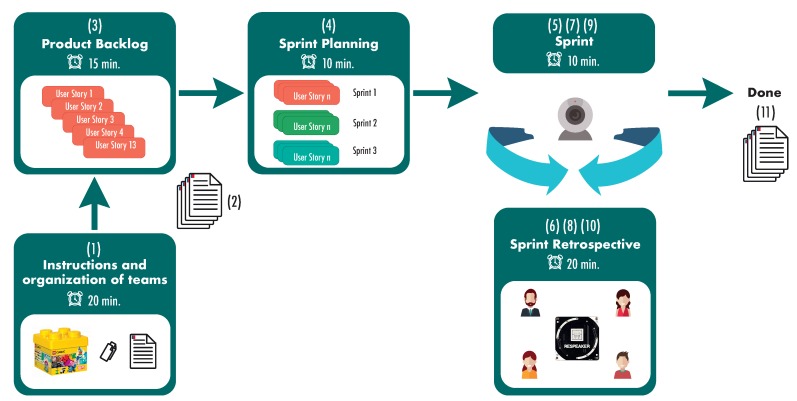
General diagram of the developed case study.

**Figure 5 sensors-19-03291-f005:**
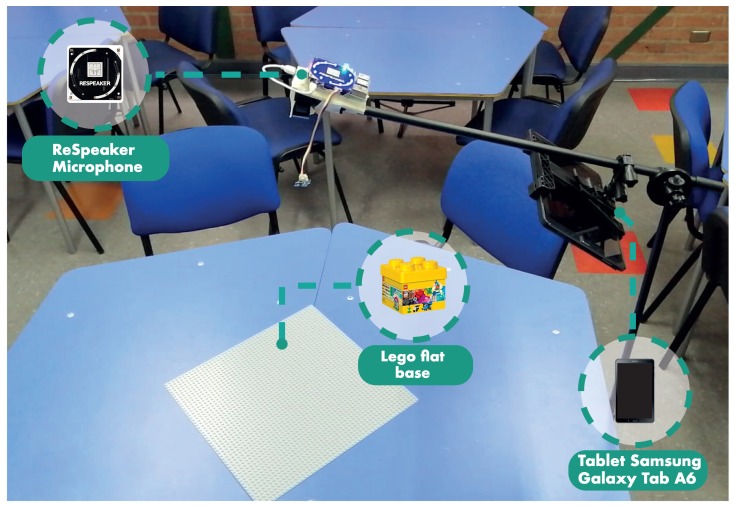
Work area for teams.

**Figure 6 sensors-19-03291-f006:**
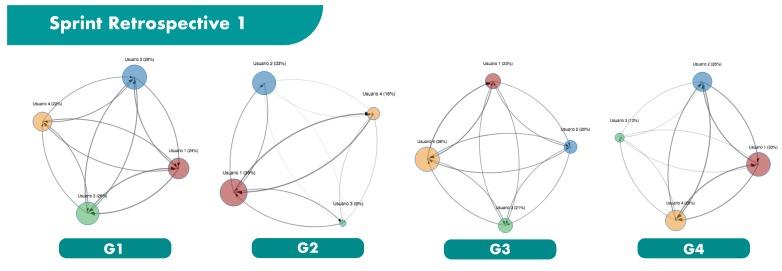
Sprint Retrospective 1.

**Figure 7 sensors-19-03291-f007:**
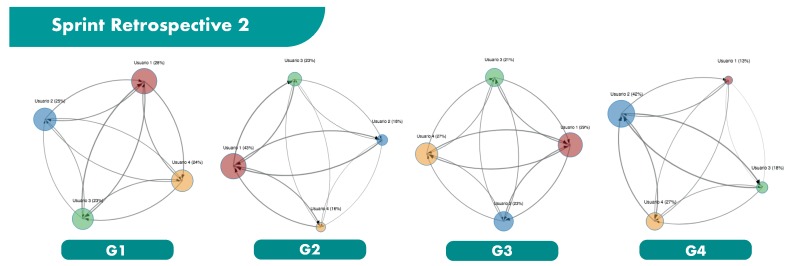
Sprint Retrospective 2.

**Figure 8 sensors-19-03291-f008:**
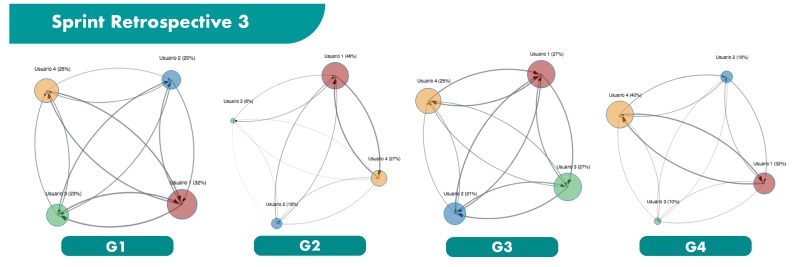
Sprint Retrospective 3.

**Figure 9 sensors-19-03291-f009:**
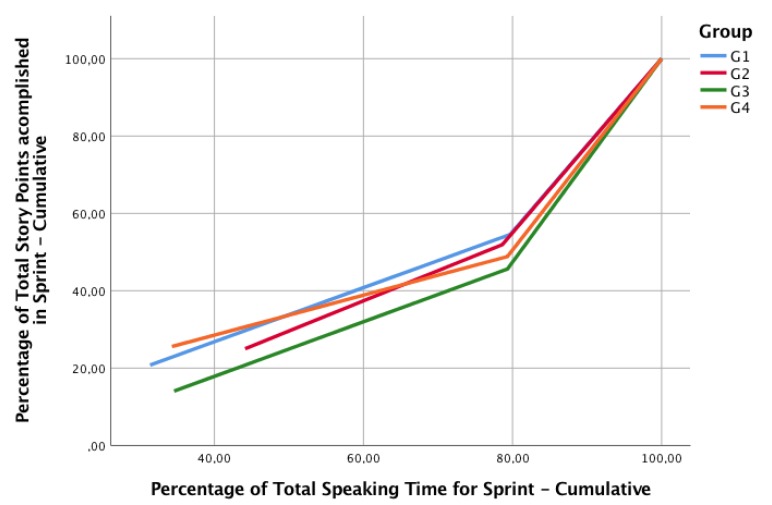
Communication in Cumulative Percentage of Speaking Time and Cumulative Percentage of Productivity in Delivered Story Points.

**Figure 10 sensors-19-03291-f010:**
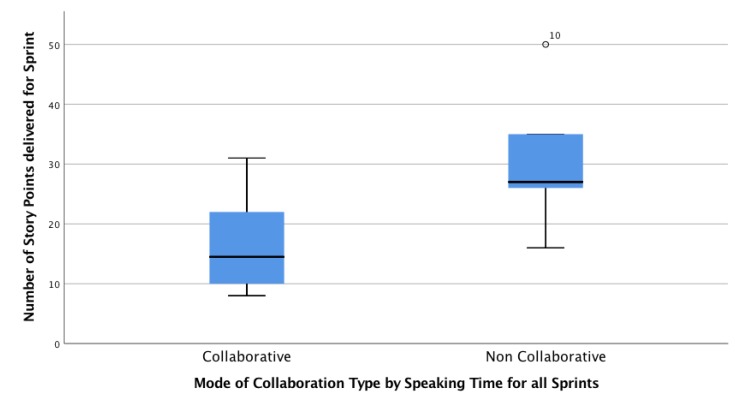
Delivered Story Points productivity per Predominant Collaboration Type by Permanence (Speaking Time).

**Figure 11 sensors-19-03291-f011:**
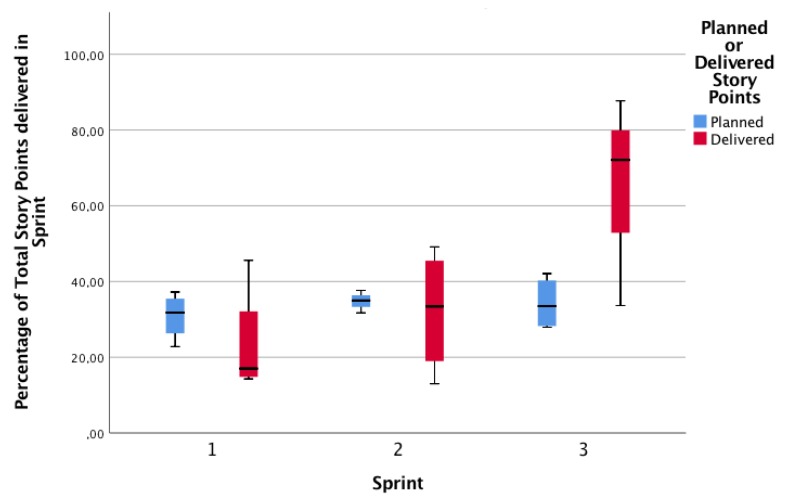
Planned vs. Delivered Story Points per Sprint.

**Figure 12 sensors-19-03291-f012:**
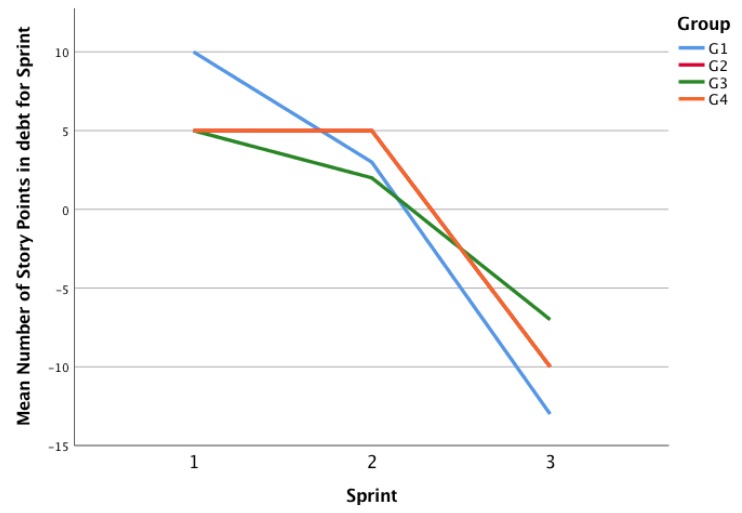
Planned vs. Delivered Story Points per sprint.

**Figure 13 sensors-19-03291-f013:**
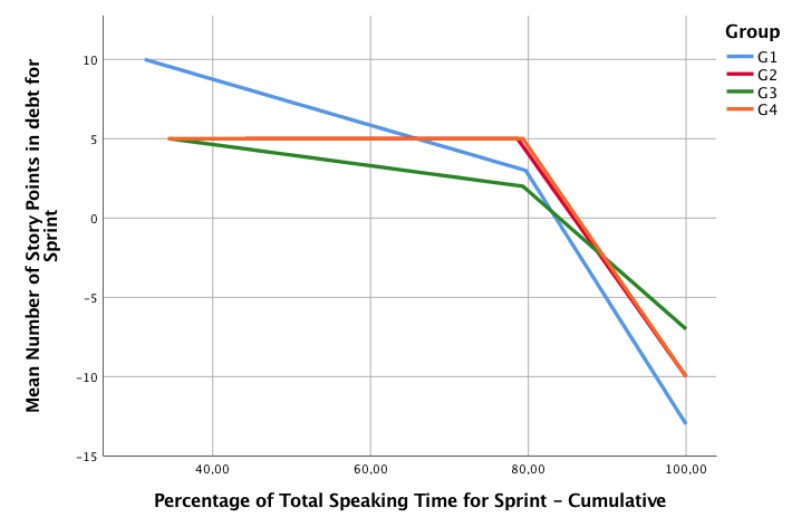
Planned vs. Delivered Story Points per Sprint.

**Table 1 sensors-19-03291-t001:** Description factor DiSC^®^.

Factor	Profile	Behavior
D—“Dominance”	Its priority is to obtain immediate results, act quickly and question others and likewise about its effectiveness. It is motivated by power, having authority, control, and success. They have a lot of confidence in themselves, they speak with frankness and forcefulness; however, they do not care about others, they are impatient and insensitive.	Direct Results-oriented Strong Tenacious Convincing
I—“Influence”	Its priority is to express enthusiasm, take action, and promote collaboration. It is motivated by social recognition, group activities, and friendly relations. They tend to be enthusiastic, sociable, optimistic, and talkative.	Extrovert Enthusiast Optimistic Vivacious Lively
S—“Steadiness”	Its priority is to support, balance, and enjoy the collaboration. They are motivated by stable environments, sincere appreciation, cooperation and opportunities to help. Usually patient, good team player, humble and good listener.	Serene Condescending Patient Humble Diplomatic
C—“Compliance”	Its priority is to ensure accuracy, balance, and challenge assumptions. He is motivated by the opportunity to use experience or increase knowledge, in addition to attending quality. They are usually meticulous, analytical, skeptical and silent	Analytical Prudent Meticulous Reserved Systemic

**Table 2 sensors-19-03291-t002:** Structure for Scrum retrospectives

Themes	Technique	Goal	Description	Time
Step 1: Set Scenario	Proud—Grateful —Learned	Help the team create an environment of positive feelings. It is a technique that helps break the ice and generate a space of trust for the team	Each member must answer the following questions in front of the group: What have you achieved in this sprint that makes you feel proud? Who from the team would you appreciate for what was done in this sprint? What have you learned in this sprint?	3 min.
Step 2: Get data, Generate ideas, Decide what to do	More—less—keep —stop—start	Help the team analyze their work process by evaluating different aspects of it.	Everyone is asked to propose ideas for changes in the process based on simple questions such as: what else should we do? What less ...? Should we keep ...?	15 min
Step 3: Close	Fun vs. Utility	Measure the mood of the team after the ceremony.	Participants are asked to mark their name in the sector that represents their feelings about the time invested. This technique helps us express how each member feels he spent his time at the meeting.	2 min.

**Table 3 sensors-19-03291-t003:** Description of user stories used.

Number	Title	As...I want...For	Validation Rules
1	Tractor	As a home builder, I want to have a tractor so I can move easily.	The rear wheels must be larger than the front wheels.
2	Tractor’s garage	As the tractor owner, I want a garage where you can store the tractor.	It must be wide and roofed
3	House with front garden.	As a citizen, I want to have a house with a front garden to enjoy the sun in summer	This house should be near the bus stop. The garden must be surrounded by a white fence.
4	Bridge	As mayor, I want a bridge so that pedestrians and vehicles can cross the river that divides the city.	The river is not large but divides the city in two. The river must have container walls.
5	Kiosk	As mayor, I want a kiosk so that citizens can relax, chat with friends or have a coffee.	It must be located near the bus stop. Must have a table and chairs outside for clients.
6	crane tower	As a home builder, I want to have a tower crane to easily transport construction materials.	The crane must be stable and located near the tractor garage. The crane must reach the roof of a 2-story building.
7	Extendable House Model	As a home builder, I want to have a house design that allows adding new parts or floors to the house	It should be possible to add a room or floors without changing the original structure of the house. The floors should follow the initial design of colors and shapes of the house.
8	Bus stop	As a citizen, I want a covered bus stop with seats so that in bad weather, it is comfortable to wait for the bus.	The stop must have spaces for advertising posters
9	Monument	As Mayor, I want a great monument to make it a point of reference in the city	The monument must be in the center of the city. It must be visible from anywhere in the city. It should be located in a green area with plants.
10	Public road	As Mayor, I want the city to have a single road that passes close to each construction.	The road must go through the Bus Stop. The road must be no more than 5 centimeters from each construction.
11	Public Hospital	As Mayor, I want the city to have a public hospital for urgent and scheduled care.	The hospital will be two floors. The hospital must have two entrances, one for emergency care and the other for scheduled care.
12	Mall	As an investor, I want to build a mall, to cover diverse needs of the citizens in a single shopping center.	The mall must have three levels.
13	Pedestrian crossing in height	As mayor, I want between the hospital and the mall a bridge over height, so that citizens can move easily to buy what they require for hospital patients.	The bridge must take care of the aesthetics of the mall and the Hospital.

**Table 4 sensors-19-03291-t004:** Number of students classified according DiSC^®^  factor.

Team	Natural Profile					Adapted Profile			
	D Dominance	I Influence	S Steadiness	C Compliance		D Dominance	I Influence	S Steadiness	C Compliance
1	0	2	1	1		0	0	3	1
2	0	1	3	0		0	1	0	3
3	2	0	1	1		0	2	1	1
4	0	0	3	0		0	0	0	3

**Table 5 sensors-19-03291-t005:** Analysis on estimation of story points.

	Story Points for User story														Descriptive Statistics		
Group	1	2	3	4	5	6	7	8	9	10	11	12	13		Amount	Average	Sd
1	1	2	5	5	8	5	8	3	5	8	13	13	1		77	5.9	4.0
2	1	5	8	8	2	13	13	5	5	13	13	5	13		104	8.0	4.5
3	1	2	5	2	2	8	8	3	3	5	8	5	5		57	4.4	2.5
4	2	3	2	3	5	3	2	3	5	5	3	5	2		43	3.3	1.3

**Table 6 sensors-19-03291-t006:** For each group (G) and sprint (S), the type of group (“NC”: non-collaborative; “C”: collaborative) is calculated from the speaking time (in seconds), according to the percentile classification under permanence criterion.

		Speaking Time (s)					Percentile				Classification				Type of Group	
G	S	*f*(1)	*f*(2)	*f*(3)	*f*(4)	Total	1	2	3	4	1	2	3	4	#*max*	Type
	1	175.4	199.1	193.0	181.9	749.3	0.47	0.67	0.60	0.53	2	3	2	2	1	NC
1	2	320.5	277.1	269.6	287.2	1154.5	0.80	0.53	0.40	0.60	3	2	2	2	1	NC
	3	158.6	95.8	110.1	120.8	485.3	0.80	0.27	0.40	0.53	3	1	2	2	1	NC
	1	469.7	404.0	111.0	225.6	1210.3	1.00	0.80	0.00	0.20	3	3	1	1	2	C
2	2	400.6	162.5	225.5	158.4	947.1	0.93	0.13	0.47	0.07	3	1	2	1	1	NC
	3	266.3	112.3	48.8	158.9	586.3	1.00	0.20	0.00	0.60	3	1	1	2	1	NC
	1	205.1	185.8	197.7	340.8	929.3	0.40	0.27	0.33	0.87	2	1	1	3	1	NC
3	2	334.5	279.7	259.8	328.5	1202.5	0.87	0.33	0.27	0.73	3	1	1	3	2	C
	3	152.1	117.7	150.8	135.0	555.6	0.73	0.33	0.67	0.47	3	1	3	2	2	C
	1	149.5	278.3	120.2	329.0	877.0	0.13	0.73	0.07	0.93	1	3	1	3	2	C
4	2	137.8	488.5	222.3	302.4	1151.0	0.00	1.00	0.20	0.67	1	3	1	3	2	C
	3	183.6	92.1	57.0	196.2	528.9	0.87	0.13	0.07	0.93	3	1	1	3	2	C

**Table 7 sensors-19-03291-t007:** For each group (G) and sprint (S), the type of group (“NC”: non-collaborative; “C”: collaborative) is calculated from the number of interventions, according to the percentile classification under prompting criterion.

		# Interventions					Percentile				Classification				Type of Group	
G	S	w(1)	w(2)	w(3)	w(4)	Total	1	2	3	4	1	2	3	4	#*max*	Type
	1	237	217	232	192	878	0.60	0.40	0.47	0.20	2	2	2	1	3	C
1	2	363	275	327	308	1273	0.80	0.13	0.73	0.53	3	1	3	2	2	C
	3	163	121	144	123	551	0.80	0.27	0.60	0.40	3	1	2	2	1	NC
	1	451	347	201	248	1247	1.00	0.67	0.07	0.13	3	3	1	1	2	C
2	2	350	229	233	201	1013	1.00	0.20	0.27	0.07	3	1	1	1	1	NC
	3	241	119	89	184	633	1.00	0.13	0.00	0.67	3	1	1	3	2	C
	1	227	195	193	242	857	0.53	0.33	0.27	0.80	2	1	1	3	1	NC
3	2	385	307	308	332	1332	0.87	0.33	0.40	0.60	3	1	2	2	1	NC
	3	200	170	168	145	683	0.73	0.53	0.47	0.20	3	2	2	1	1	NC
	1	76	75	36	81	268	0.87	0.73	0.00	0.93	3	3	1	3	3	C
4	2	107	207	143	152	609	0.00	0.93	0.47	0.67	1	3	2	3	2	C
	3	114	81	54	115	364	0.87	0.33	0.07	0.93	3	1	1	3	2	C

**Table 8 sensors-19-03291-t008:** Global metrics for Groups (G): Collaboration Type by Permanence (CTPer), Collaboration Type by Prompting (CTProm), total Speaking Time (ST), total Number of Interventions (NI), total Planned Story Points (SP-P), total Delivered Story Points (SP-D), Predominant Leadership Style (PLS), Predominant Personality Type (PPT).

G	CTPer	CTProm	ST	NI	SP-P	Total SP-D	PLS	PPT
1	NC	C	2389.06	2702	77	77	Affiliative	Influenced
2	NC	C	2743.64	2893	104	104	Undefined	Steady
3	C	NC	2687.42	2872	57	57	Democratic	Dominant
4	C	C	2556.88	1241	43	43	Democratic	Steady

**Table 9 sensors-19-03291-t009:** Sprint metrics for Groups (G), Sprint (S), Speaking Time for the Sprint (ST), Percentage of Speaking Time (ST%), Cumulative Speaking Time Percentage (ST%C), Collaboration Type by Permanence (CTPerm), Number of Interventions (NI), Percentage of Number of Interventions (NI%), Cumulative Percentage of Number of Interventions (NI%C), Collaboration Type by Prompting (CTProm), Planned Story Points (SP-P), Delivered Story Points (SP-D), Percentage of Delivered Story Points (SP-D%), Cumulative Percentage of Story Points Delivered (SP-D%C), Story Points Debt (SPDebt).

G	S	ST	ST%	ST%C	CTPer	NI	NI%	NI%C	CTPRom	SP-P	SP-D	SP-D%	SP-D%C	SPDebt
1	1	749.32	31.36	31.36	NC	878	32.49	32.49	C	26	16	20.78	20.78	10
2	1	1210.26	44.11	44.11	C	1247	43.10	43.10	C	31	26	25.00	25.00	5
3	1	929.32	34.58	34.58	NC	857	29.84	29.84	NC	13	8	14.04	4.04	5
4	1	876.98	34.30	34.30	C	268	21.60	21.60	C	16	11	25.58	25.58	5
1	2	1154.48	48.32	79.68	NC	1273	47.11	79.60	C	29	26	33.77	54.55	3
2	2	947.06	34.52	78.63	NC	1013	35.02	78.12	NC	33	28	26.92	51.92	5
3	2	1202.48	44.74	79.32	C	1332	46.38	76.22	NC	20	18	31.58	45.62	2
4	2	1151.00	45.02	79.32	NC	609	49.07	70.67	C	15	10	23.26	48.84	5
1	3	485.26	20.31	100.00	NC	551	20.39	100.00	NC	22	35	45.45	100.00	−13
2	3	586.32	21.37	100.00	NC	633	21.88	100.00	C	40	50	48.08	100.00	−10
3	3	555.62	20.67	100.00	C	683	23.78	100.00	NC	24	31	54.39	100.00	−7
4	3	528.90	20.69	100.00	C	364	29.33	100.00	C	12	22	51.16	100.00	−10

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
