# Peer review of "Introducing Low-Cost Sensors into the Classroom Settings: Improving the Assessment in Agile Practices with Multimodal Learning Analytics"

_sensors, 2019, doi:10.3390/s19153291_

Round 1

Reviewer 1 Report

The authors present a mature and robust paper that frames multimodal learning analytics (MMLA) into a tangible learning activity for agile practices. The paper is clearly organized, well argued, and well written. The authors present a strong case and novel area of investigation. The research design is built up from several publications that bring the threads together into a robust intervention, that captures qualitative and quantitative research instruments and demonstrates how MMLA can be utilized as a part of the researcher's toolbox. The research design is clear are tied to the hypotheses; however, more clarity about the number of participants is needed. One issue that the authors should consider is arguing more for the low-cost nature of the Re-Speaker / Pi set up and what future pragmatic approaches could be

Additionally, suggest the future plans include more substantial scale testing since  case participants were limited to four groups. In addition  there could be a need to briefly discuss how the analysis of the qualitative could be automated since at the moment the cases are heavily research oriented.

From a minor editing perspective several small paragraphs need to be reformed (lines 50-59, 147-154). Additionally, lines 172-173 have extra space for the ending period.

Author Response

First of all, we want to thank the reviewers for their reviews and useful advice. We are in agreement with all of their observations. This is why, below, we indicate how the observations were addressed in the new version of the article. We hope we have responded to each observation.

Reviewer 2 Report

This paper presents an exploratory study about Multimodal Learning Analysis applications as a complement to the evaluation of different skills such as collaboration and communication. The paper is well written and steps for the evaluation are detailed. The references are suitable and updated.

The major drawback of this study is the lack of direct results and discussion about the introduction of low-cost Sensors into the Classroom Settings. It is not clear the importance of these sensors in the evaluation process. The authors should enhance this analysis in discussion in order to better explain the paper in the Sensors journal’s scope.

Author Response

(The authors gave the same response as above.)
